

# Radionuclide contamination in flood sediment deposits in the coastal rivers draining the main radioactive pollution plume of Fukushima Prefecture, Japan (2011–2020)

Olivier Evrard[1], Caroline Chartin[1,2], J. Patrick Laceby[1,3], Yuichi Onda[4], Yoshifumi Wakiyama[5], Atsushi Nakao[6], Olivier Cerdan[7], Hugo Lepage[1,8], Hugo Jaegler[1,8], Rosalie Vandromme[7], Irène Lefèvre[1], Philippe Bonté[1]

[1] Laboratoire des Sciences du Climat et de l'Environnement (LSCE/IPSL), Unité Mixte de Recherche 8212 (CEA/CNRS/UVSQ), Université Paris-Saclay, Gif-sur-Yvette, France
[2] Georges Lemaître Centre for Earth and Climate Research, Earth and Life Institute, Université catholique de Louvain, Louvain-la-Neuve, Belgium
[3] Alberta Environment and Parks (AEP), Calgary, Alberta, Canada
[4] Centre for Research in Isotopes and Environmental Dynamics (CRIED), Tsukuba University, Tsukuba, Japan
[5] Institute of Environmental Radioactivity (IER), University of Fukushima, Fukushima, Japan
[6] Graduate School of Life and Environmental Sciences, Kyoto Prefectural University, Kyoto, Japan
[7] Bureau de Recherches Géologiques et Minières (BRGM), Département Risques et Prévention, Orléans, France
[8] Institut de Radioprotection et de Sûreté Nucléaire (IRSN), PSE-ENV, SRTE/LRTA, SAME/LERCA, BP 3, 13115, Saint-Paul-lez-Durance, France

*Correspondence to*: Olivier Evrard (olivier.evrard@lsce.ipsl.fr).

**Abstract.** Artificial radionuclides including radiocesium ($^{134}Cs$ and $^{137}Cs$) and radiosilver ($^{110m}Ag$) were released into the environment following the Fukushima Dai-ichi Nuclear Power Plant accident in March 2011. These particle-bound substances deposited on soils of Northeastern Japan located predominantly within a ~3000 km² radioactive fallout plume and drained by several coastal rivers to the Pacific Ocean. The current dataset that can be accessed at https://doi.pangaea.de/10.1594/PANGAEA.928594 compiles gamma-emitting artificial radionuclide activities measured in 782 sediment samples collected from 27 to 71 locations during 16 fieldwork campaigns conducted in Japan between November 2011 and November 2020 in river catchments draining the main radioactive plume. This database may be useful to evaluate and anticipate the post-accidental redistribution of radionuclides in the environment and for the spatial validation of models simulating the transfer of radiocesium across continental landscapes.

## 1. Introduction

The accident that occurred in March 2011 at the Fukushima Dai-ichi Nuclear Power Plant (FDNPP) released large quantities of radionuclides into the environment (Leelossy et al., 2011). Among the radioactive substances emitted, two radiocesium isotopes ($^{134}Cs$ and $^{137}Cs$) are the most problematic over the medium to long term, as they were released in abundant quantities (Shozugawa et al., 2012). Furthermore, they are characterized by relatively long half-lives (2 years for $^{134}Cs$ and 30 years for





$^{137}$Cs), which may cause their persistence in the environment (Evrard et al., 2015). Both radiocesium isotopes were released in equivalent proportions during the accident, with initial $^{134}$Cs:$^{137}$Cs activity ratios close to 1 (Kobayashi et al., 2017). Other artificial radionuclides such as silver-110 metastable ($^{110m}$Ag) – an activation product of $^{109}$Ag found in control rods or in the

alloy used to seal the head of the reactor – were found in the environment in the vicinity of FDNPP (Le Petit et al., 2012). Radiocesium deposition on soils of the Fukushima Prefecture generated the formation of a ~3000 km² radioactive pollution plume extending to the northwest of the FDNPP (Yasunari et al., 2011). Radiocesium and radiosilver were shown to have a strong affinity for the fine mineral fractions (typically the clay- and silt-sized fractions) of these soils (Fan et al., 2014;Lepage et al., 2014;Nakao et al., 2014). However, these substances have also been found in coarser particle fractions including sand-

sized material (Tanaka et al., 2014). As the soils located in the main radioactive pollution plume are drained by several coastal river systems to the Pacific Ocean, the redistribution of the initial contamination through water erosion and riverine sediment transfer processes was anticipated (Evrard et al., 2015;Onda et al., 2020).

This region of Northeastern Japan is exposed to frequent heavy rainfall events including typhoons occurring mainly during summer and early in autumn (Laceby et al., 2016a). These events often generate extensive soil erosion and flooding in the

region (Evrard et al., 2020). They therefore provide a significant mechanism redistributing the radioactive contamination from the soils exposed to the initial fallout to the Pacific Ocean, after transiting these coastal river systems (Nagao et al., 2013;Evrard et al., 2014).

Monitoring the spatial and the temporal redistribution of this radioactive contamination transported with sediment during the post-accident period is therefore crucial to improve our understanding of soil erosion and sediment transfer processes in rivers

exposed to frequent typhoons and to evaluate the effectiveness of remediation measures. In Fukushima, decontamination mainly consisted in removing the topsoil layer concentrating fallout radionuclides and in replacing it with a new substrate devoid of artificial radionuclides (Evrard et al., 2019c).

To conduct this spatial and temporal monitoring, sediment samples were collected systematically at various locations in the coastal catchments draining the main radioactive pollution plume to characterize the evolution of the radioactive contamination

transiting these rivers between 2011 and 2020. The data described here was partly used in previous publications to investigate the potential changes in sources (i.e. soil types(Lepage et al., 2016), land use types(Laceby et al., 2016b;Evrard et al., 2019b;Evrard et al., 2019a;Huon et al., 2018), and surface material vs. subsoil(Evrard et al., 2016)) supplying material to the rivers. The current database provides the gamma-emitting radionuclide activities measured in these sediment deposits along with the radioactive dose rates measured in the river channel and in the nearby soils during these surveys. This dataset provides

a unique and uniform data compilation collected using consistent sampling and internationally-calibrated gamma analyses methods throughout the first decade (2011 – 2020) that followed the FDNPP accident. It provides a useful complement to the 6-year dataset (2011–2017) of radiocesium fluxes analysed in sediment draining the Fukushima radioactive plume (Taniguchi et al., 2020).






## 2. Dataset

The dataset includes the following fields for each record:

| | | |
|---|---|---|
| | LABEL | Sample name |
| 75 | SAMPLING DATE | Sampling date used as reference date for radionuclide decay-correction |
| | ANALYSIS DATE | Date of gamma spectrometry analysis |
| | CAMPAIGN | Number of fieldwork campaign (from 1 to 16; campaign #12 was cancelled due to flood occurrence) |
| | RIVER | Name of the river catchment where the sample was collected |
| 80 | LATITUDE | Latitude (WGS 1984) |
| | LONGITUDE | Longitude (WGS 1984) |
| | DOSE RATE SOIL | Radioactive dose rate measured on soils nearby the river ($\mu$Sv h$^{-1}$) |
| | DOSE RATE RIVER | Radioactive dose rate measured on recent sediment deposits in the river channel ($\mu$Sv h$^{-1}$) |
| | CS-137 | $^{137}$Cs concentration analysed by gamma spectrometry (Bq kg$^{-1}$) |
| 85 | CS-134 | $^{134}$Cs concentration analysed by gamma spectrometry (Bq kg$^{-1}$) |
| | AG-110M | $^{110m}$Ag concentration analysed by gamma spectrometry (Bq kg$^{-1}$) |

## 3. Methods

Sixteen sediment sampling campaigns (numbered from 1 to 16) were organized between November 2011 and November 2020. Sampling occurred bi-annually (i.e. in autumn after the typhoon season, and in spring, after the snowmelt runoff) between
November 2011 and November 2016. Then, the campaigns occurred after the typhoon season late in October or early in November between 2017 and 2020 at 27 to 71 locations. Sediment could not be collected during campaign # 12 (Spring 2017) because of the occurrence of a flood leading to the resuspension of sediment in the water column during that period.

In total, 782 sediment samples were collected in river catchments draining the main radioactive plume (Figure 1). Fine sediment samples were taken from material deposited after the last major event at the same sites, during each of the fifteen
campaigns. These lag deposit samples were comprised of fine particulate material that settled on channel banks, inset benches and floodplains during the falling limb of the last significant hydro-sedimentary event. Ten subsamples (~5 g per subsample) of recently deposited material were taken with a plastic spatula over a 5 m reach and composited into one sample. In the field, radiation dose rates were systematically measured at 5-cm height using a radiameter (LB123 D-H10, Berthold Technologies) in recent sediment drape deposits and in nearby soils along rivers (i.e. in a 10 to 20 m wide area along rivers).

All samples were dried at 40°C for ~48 h, sieved to 2 mm, disaggregated, and pressed into 15 mL polyethylene containers for analysis. Gamma-emitting radionuclide activities were determined by gamma spectrometry using low-background coaxial HyperPure Germanium detectors (Canberra/Ortec). Samples were analysed for 30,000 to 200,000 s. $^{137}$Cs activities were





measured at the 662 keV emission peak. $^{134}$Cs activities were calculated as the mean of activities measured at both 604 keV and 795 keV emission peaks. Although the presence of $^{110m}$Ag in a sample was confirmed when peaks were detected at 885,

937 and 1384 keV simultaneously, the activities in this radionuclide were calculated from the 885-keV peak only. All radionuclide activities were decay-corrected to the sampling date. Errors reached ca. 5-10% on $^{134}$Cs and $^{137}$Cs activities, and ca. 15-20% on $^{110m}$Ag activities at the 95% confidence level. All measured counts were corrected for background levels measured at least every 2 months as well as for detector and geometry efficiencies. Results were systematically expressed in Bq kg$^{-1}$ of dry weight. Quality assurance was conducted using certified International Atomic Energy Agency (IAEA) reference

materials (i.e. IAEA-444, IAEA-375) as well as a multi-gamma resin produced by IRSN (Institut de Radioprotection et de Sûreté Nucléaire, France) with elevated $^{137}$Cs (160,000 Bq kg$^{-1}$; reference date: 7 January 2011) and $^{134}$Cs activities (330,000 Bq kg$^{-1}$) prepared in the same containers as the samples.

## 4.    Results

Overall, $^{137}$Cs activities decreased by 93% in sediment during the monitoring period, from a mean of 28,516 Bq kg$^{-1}$ (range:

126 – 715,647 Bq kg$^{-1}$) in 2011 to a mean of 2115 Bq kg$^{-1}$ in 2020 (range: 68 – 11,928 Bq kg$^{-1}$; Figure 2). This decreasing trend is consistent despite the strong spatial variations due to the heterogeneity of the initial radioactive contamination levels (Figure 1). The decrease in $^{137}$Cs activities throughout time mainly occurred after the occurrence of typhoon Etau in 2015, with mean $^{137}$Cs activities declining from 20,397 Bq kg$^{-1}$ in November 2015 to 3419 Bq kg$^{-1}$ in June 2016. This period also coincided with the occurrence of widespread decontamination works in the region (Evrard et al., 2016).

During the 2011–2020 monitoring period, $^{134}$Cs activities in sediment decayed rapidly, with $^{134}$Cs:$^{137}$Cs activity ratios decreasing from a mean of 0.66 (standard deviation – SD: 0.04) in November 2011, to a mean of 0.05 (SD: 0.01) in November 2020 (Figure 3).

Because of an even more rapid radioactive decay (half-life of ~250 days), $^{110m}$Ag activities in sediment were no longer detected after May 2013. This absence of detection from 2013 onwards is also due to the emission of $^{110m}$Ag in much lower abundance

compared to $^{137}$Cs. Indeed, $^{110m}$Ag:$^{137}$Cs activity ratios between 0.002–0.010 (decay-corrected to March 2011) were measured in soils and sediment of the main plume, which could be used during the early post-accidental conditions (2011–2012) to investigate the dispersion of the radioactive contamination in the coastal rivers (Lepage et al., 2014).

The decrease in radioactive contamination with time measured in the sediment was also reflected by a strong decrease in the radioactive decrease of the ambient dose rates (Figure 4), from a mean of 3.3 μSv h$^{-1}$ (range: 0.1–40 μSv h$^{-1}$) in November

2011 to a mean of 0.5 μSv h$^{-1}$ (range: 0.1–2.6 μSv h$^{-1}$). From 2019 onwards, the radiation dose rates emitted by recent sediment deposits and nearby soils could no longer be distinguished from the general background signal, and measurements were therefore no longer conducted during the last two fieldwork campaigns in 2019 and 2020. The strong decrease in $^{137}$Cs activities observed after the occurrence of typhoon Etau in 2015 (Figure 2) is also reflected by a significant decline in radioactive dose rates measured on recent sediment deposits (mean of 0.6 μSv h$^{-1}$ in June 2016 compared to a mean of 1.3 μSv h$^{-1}$ in November

2015; Figure 4).



## 5. Data availability

The data is archived at https://doi.pangaea.de/10.1594/PANGAEA.928594.

## 6. Conclusions

This database compiles radiocesium (and radiosilver during the early post-accidental stage) activities analysed in recent sediment deposits collected following a homogeneous protocol in the coastal catchments draining the main radioactive pollution plume in the Fukushima Prefecture, Japan. These results demonstrate that the radiocesium levels in sediment transiting these rivers decreased by more than 90% between 2011–2020. This is confirmed by the similar decline (~85%) in
radioactive dose rates observed in the field between 2011–2018. This dataset demonstrates the impact of the rapid decontamination of catchments exposed to accidental radioactive fallout in less than a decade, which will be useful for model output validation and for anticipating the fate of residual radionuclides in the environment. In the future, monitoring of radiocesium activities will continue in Fukushima coastal rivers in order to investigate the impact of recultivation of decontaminated areas on these transfers (Bourdet, 2021).


**Author contributions**

All the authors participated to (at least part of) the sixteen sediment sampling campaigns (and the associated radioactive dose rate measurements), as well as to sample preparation in the laboratory. I.L. and P.B. conducted the gamma spectrometry analyses and the associated quality control. O.E. wrote the manuscript, and all co-authors provided feedback and revised the
text. All the authors declare that this database is an original product of their collaborative work conducted in Fukushima coastal catchments since 2011. Although interpretations based on some of the data presented in the current manuscript have been published in previous publications of the group (see the references cited in the Introduction), the objective of the current data paper was to provide a unique and uniform database compiling all the radioactive dose rates and radionuclide concentrations measured by the Franco-Japanese consortium in sediment transiting coastal rivers draining the main radioactive plume during
the first decade (2011–2020) that followed the FDNPP accident. The publication of this raw and unique data compilation should facilitate the dissemination of data acquired in these post-accidental conditions among the international community.

**Competing interests**

The authors declare that there is no conflict of interest.


**Acknowledgements**

The collection and the analysis of the sediment samples were funded by the TOFU (ANR-11-JAPN-001) and the AMORAD (ANR-11-RSNR-0002) projects, under the supervision of the French National Research Agency (ANR, Agence Nationale de la Recherche). The support of CEA (Commissariat à l'Energie Atomique et aux Energies Alternatives, France), CNRS (Centre




National de la Recherche Scientifique, France) and JSPS (Japan Society for the Promotion of Science) through the funding of PhD fellowships (H. Lepage, H. Jaegler) and collaboration projects (grant no. PRC CNRS JSPS 2019-2020, no.10; CNRS International Research Project – IRP – MITATE) is also gratefully acknowledged.

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



**Figures**

Figure 1. Location of the sediment samples collected between 2011–2020 in the river catchments draining the main radioactive plume nearby FDNPP. Background map of initial radiocesium concentrations after Chartin et al. (2013).

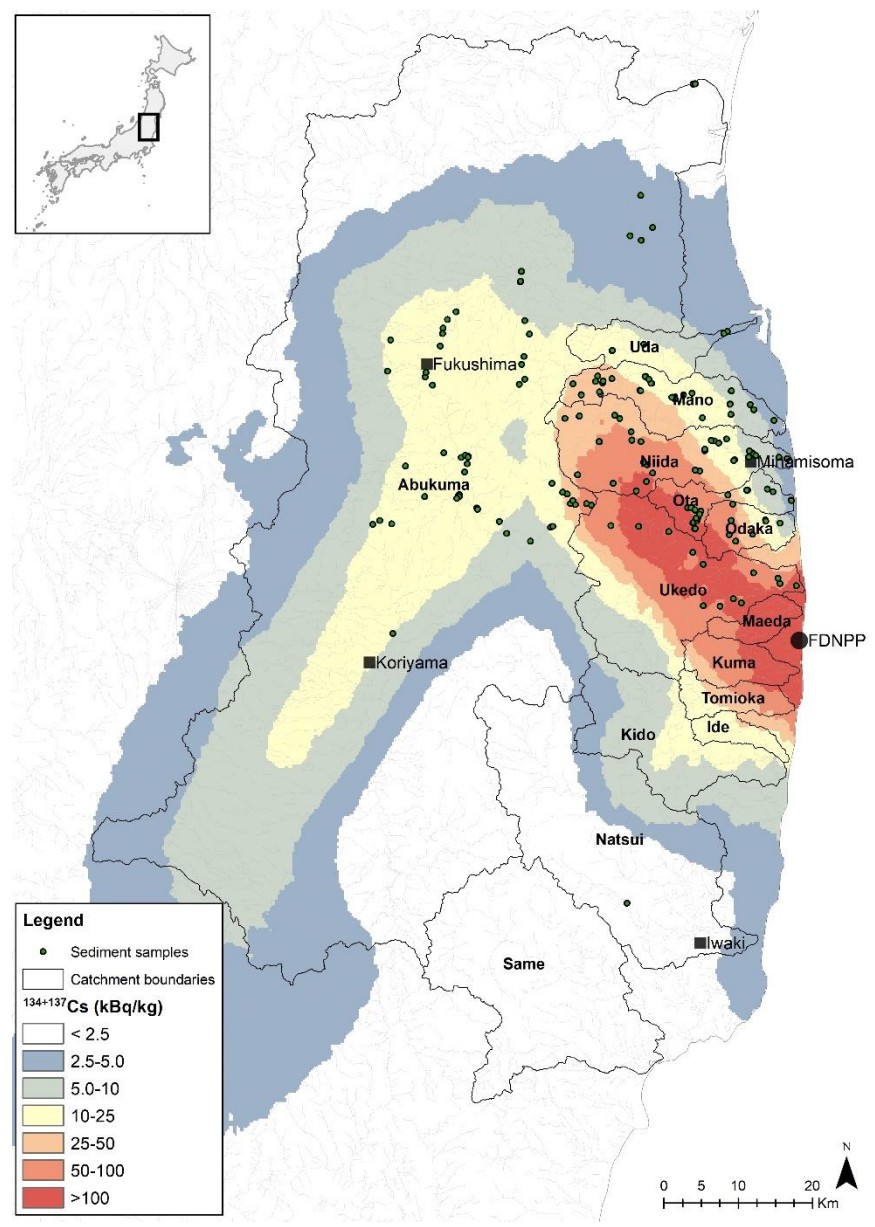






Figure 2. Evolution of $^{137}$Cs concentrations in river sediment samples collected in the river catchments draining the main radioactive plume nearby FDNPP between Nov. 2011 and Nov. 2020 (all activities were decay-corrected to the sampling date).

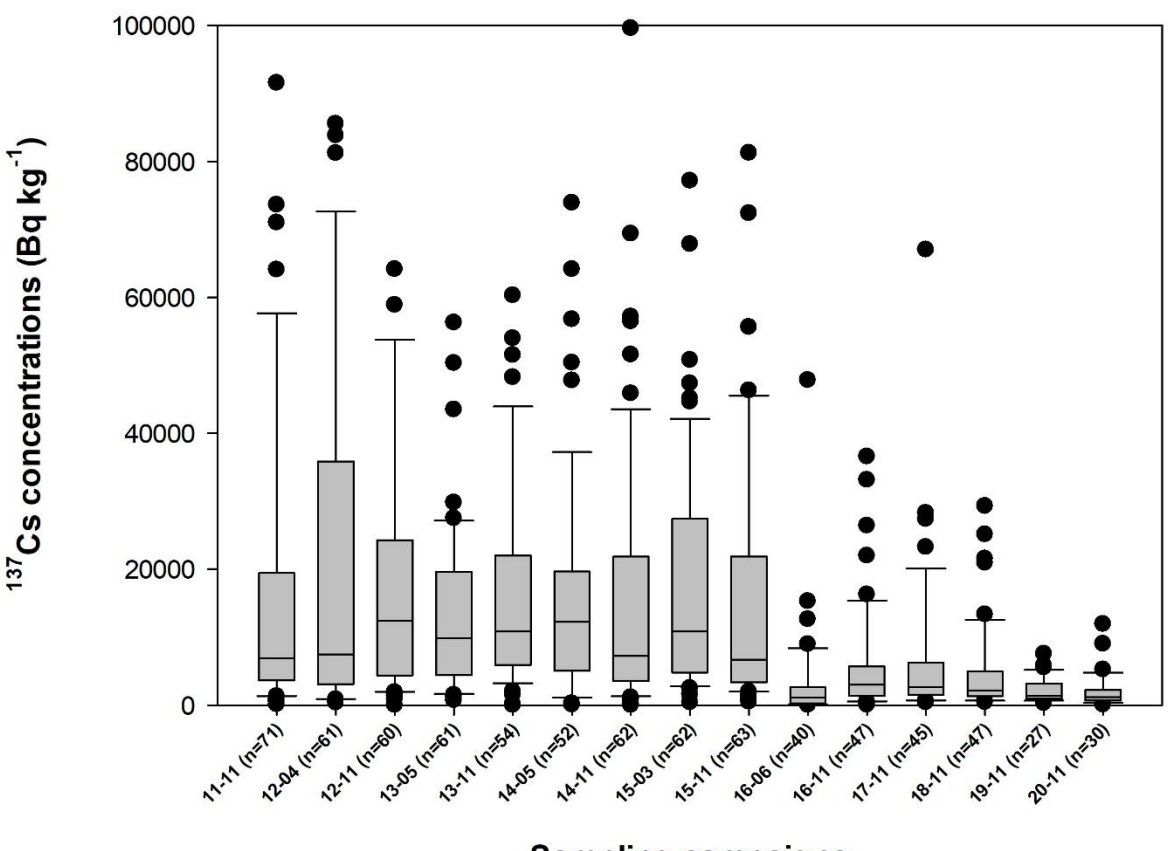



Figure 3. Evolution of $^{137}$Cs:$^{134}$Cs activity ratios in river sediment samples collected in the river catchments draining the main radioactive plume nearby FDNPP between Nov. 2011 and Nov. 2020 (all activities were decay-corrected to the sampling date).

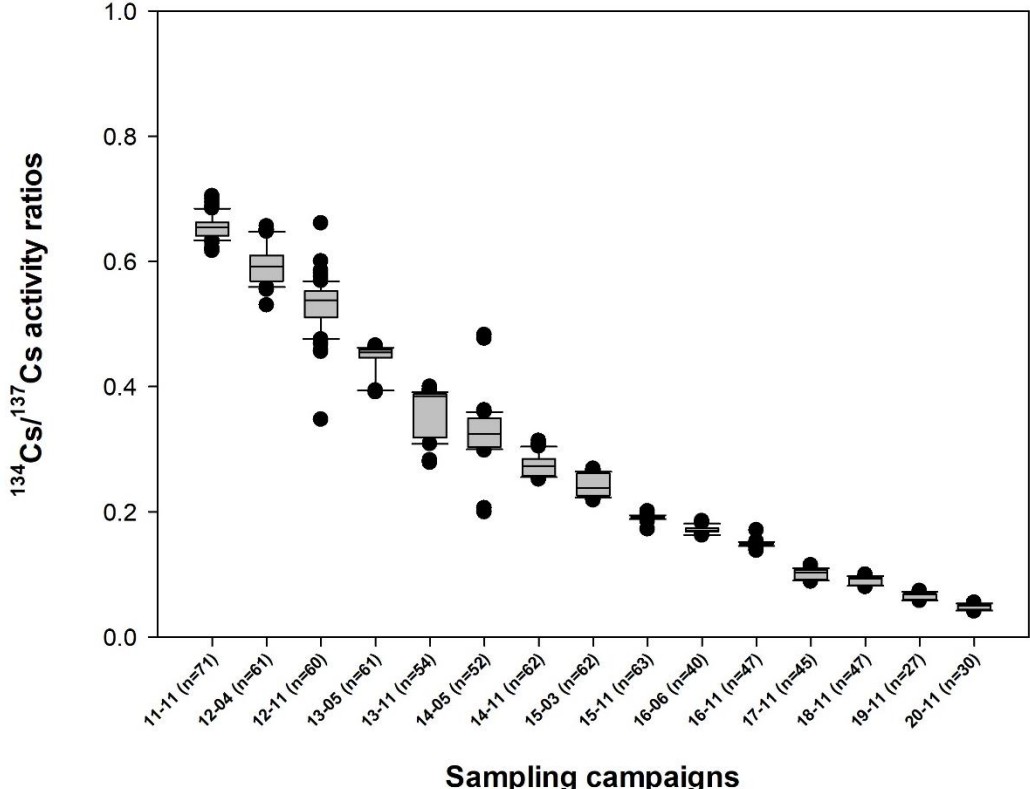





Figure 4. Evolution of radioactive dose rates on recent sediment deposits in the river catchments draining the main radioactive plume nearby FDNPP between Nov. 2011 and Nov. 2018.

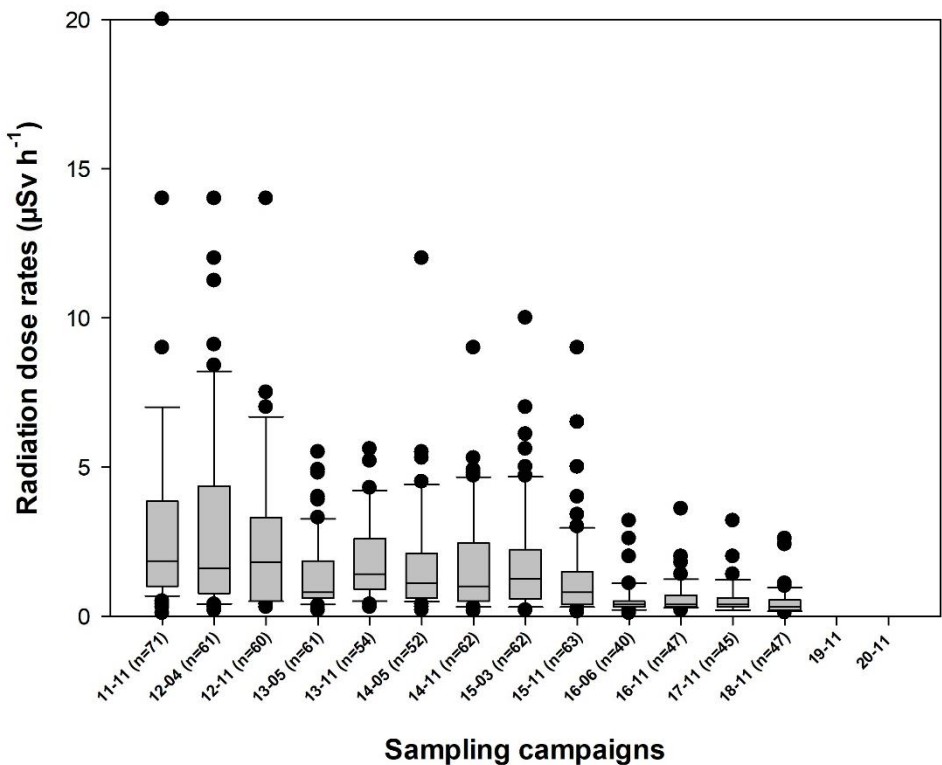