# Peer review of "Radionuclide contamination in flood sediment deposits in the coastal rivers draining the main radioactive pollution plume of Fukushima Prefecture, Japan (2011–2020)"

_Earth System Science Data, 2021_

## Author Response (AR1)

| Reviewer 1 | |
|---|---|
| **Reviewer's comments** | **Replies** |
| *General comments*
This paper shows a dataset of artificial radionuclides measured in 782 sediment samples collected from 27 to 71 locations during 16 fieldwork campaigns conducted in Japan between November 2011 and November 2020 in river catchments. This dataset is very useful to evaluate redistribution of radionuclides in the environment with time and for the spatial validation of models simulating the transfer of radiocesium caused by the base and storm flow. I think that detailed information on sampling methods is lack, therefore I recommend the manuscript should be revised according to comments. | The authors are grateful to the reviewer for providing this general positive comment and suggestions to improve the manuscript. |
| *Specific comments*
*Abstract* P1. L28. 'sediment samples' is vague. I suggest the author add the area took samples. | We added the drainage area across which samples were collected (L. 28). |
| *Dataset*
If you have measured soil characteristics such as soil density and particle size distribution, please show these data because it is very important information when sediment and radiocesium transport are simulated by river and watershed modeling. | Unfortunately, these properties could not be measured on all the samples and we therefore decided not to include this data in the current database. However, we plan to do it in a future dataset. |
| *Methods*
I suggest the author add more detailed sampling methods. I was wondering how to take samples from surface soils and sediment, and how much depth and volume you took. If this information is lack, it is difficult to use these data. | Details on the sampling method were added (LL. 96-98). |
| *Results*
I suggest the author add discussions about differences before and after typhoon Hagibis and Bualoi events in 2019 because precipitation (ex. meteorological station at Namie) during these events was much larger than during typhoon Etau in 2015. | This information has been added to the text (LL.120-128). |
| *Technical corrections*
P3. L98. Not 5-cm but 1-cm? The dataset that can be accessed at https://doi.pangaea.de/10.1594/PANGAEA.928594 shows 'Radiation dose rates were systematically measured in the field (at 1-cm height from the soil)...', but the manuscript is written 'In the field, radiation dose rates were systematically measured at 5-cm height using a radiometer...'. | We are grateful to the reviewer for catching this error, we corrected the manuscript accordingly (L. 99). |
| Reviewer 2 | |
| **Reviewer's comments** | **Replies** |
| *General comments*
This paper shows a worthful dataset about artificial radionuclide in Fukushima prefecture. Authors conducted field sampling works for a decade, and got numerous data. This dataset will be utilized for the various research relating radionuclide dynamics. I put a specific comment for the results. | The authors are grateful to the reviewer for providing this general positive comment and a specific suggestion to improve the manuscript. |

| | |
|---|---|
| *Specific comment*
In the result section, a flood event caused by typhoon Etau in 2015 is focused for the one of the reasons of decreasing $^{137}$Cs concentration. Another serious flood event occurred in October 2019. Remarkable decrease is found in the time of 19-11 and 20-11 in Figure 2. Some more explanation is required about this event. | This information has been added to the text (LL.120-128). |